# Contextualizing injury severity from occupational accident reports using an optimized deep learning prediction model

Mohamed Zul Fadhli Khairuddin[1], Suresh Sankaranarayanan[2], Khairunnisa Hasikin[3], Nasrul Anuar Abd Razak[3] and Rosidah Omar[4]

[1] Institute of Medical Science Technology, Universiti Kuala Lumpur, Kajang, Selangor, Malaysia
[2] Department of Computer Science, College of Computer Science and Information Technology, King Faisal University, Hofuf, Kingdom of Saudi Arabia
[3] Department of Biomedical Engineering, Faculty of Engineering, Universiti Malaya, Kuala Lumpur, Kuala Lumpur, Malaysia
[4] Occupational and Environmental Health Unit, Kedah State Health Department, Alor Setar, Kedah, Malaysia



Corresponding authors
Suresh Sankaranarayanan,
ssuresh@kfu.edu.sa
Khairunnisa Hasikin,
khairunnisa@um.edu.my

## ABSTRACT

**Background:** This study introduced a novel approach for predicting occupational injury severity by leveraging deep learning-based text classification techniques to analyze unstructured narratives. Unlike conventional methods that rely on structured data, our approach recognizes the richness of information within injury narrative descriptions with the aim of extracting valuable insights for improved occupational injury severity assessment.

**Methods:** Natural language processing (NLP) techniques were harnessed to preprocess the occupational injury narratives obtained from the US Occupational Safety and Health Administration (OSHA) from January 2015 to June 2023. The methodology involved meticulous preprocessing of textual narratives to standardize text and eliminate noise, followed by the innovative integration of Term Frequency-Inverse Document Frequency (TF-IDF) and Global Vector (GloVe) word embeddings for effective text representation. The proposed predictive model adopts a novel Bidirectional Long Short-Term Memory (Bi-LSTM) architecture and is further refined through model optimization, including random search hyperparameters and in-depth feature importance analysis. The optimized Bi-LSTM model has been compared and validated against other machine learning classifiers which are naïve Bayes, support vector machine, random forest, decision trees, and K-nearest neighbor.

**Results:** The proposed optimized Bi-LSTM models' superior predictability, boasted an accuracy of 0.95 for hospitalization and 0.98 for amputation cases with faster model processing times. Interestingly, the feature importance analysis revealed predictive keywords related to the causal factors of occupational injuries thereby providing valuable insights to enhance model interpretability.

**Conclusion:** Our proposed optimized Bi-LSTM model offers safety and health practitioners an effective tool to empower workplace safety proactive measures, thereby contributing to business productivity and sustainability. This study lays the foundation for further exploration of predictive analytics in the occupational safety and health domain.

# INTRODUCTION

The prioritization of workplace safety and health is crucial for both employees and employers. Ensuring a safe and healthy workplace is not only a legal and moral obligation but also, an essential determinant in sustaining job productivity and optimizing staff efficiency. Occupational injuries have the potential to result in substantial financial ramifications for the organization (*Debela, Azage & Begosaw, 2021*; *Kim & Park, 2021*), decreased employee morale as a consequence of protracted medical recuperation (*Chin et al., 2018*; *Kendrick et al., 2017*), and adverse effects on the overall quality of life within society (*Tompa et al., 2021*). Therefore, it is imperative to conduct a timely and precise evaluation of the severity of occupational injuries to implement suitable and efficient workplace safety intervention strategies. However, the evaluation of occupational injury severity can present multifaceted challenges due to its dependence on the manual assessment of occupational injury textual reports. This task is time-consuming and requires specific expertise; thus, making it susceptible to human error (*Kim & Chi, 2019*). As a result, most previous studies have focused on the use of structured categorical data for the analysis of occupational accidents (*Chadyiwa, Kagura & Stewart, 2022*; *Marucci-Wellman, Corns & Lehto, 2017*), whereas the examination of textual reports on industrial injuries has been neglected (*Abbasianjahromi & Aghakarimi, 2021*). The rapid development of big data technology has led to significant advancements in natural language processing (NLP) and Artificial Intelligence (AI) techniques, resulting in promising performance in text categorization tasks (*Sarkar et al., 2019*).

An NLP-based text mining technique is defined as the process of extracting and deriving information from unstructured text data to generate feature representations for classification and prediction analyses (*Khattak et al., 2019*). This is done through feature engineering or text representation techniques, such as text vectorizers; Bag of Words (BoW), and Term Frequency-Inverse Document Frequency (TF-IDF), as well as, the word embeddings pre-trained model, such as Word2Vec and Global Vector (GloVe). Both text vectorizers, BoW and TF-IDF are easily executed and compatible (*Pan et al., 2020*); however, they do not define semantic relationships in context (*Goldberg, 2022*). To overcome this limitation, the word embeddings approach is recommended as it is capable of preserving the relationship of semantic and syntactic linguistics in text documents (*Young et al., 2018*).

Most recent studies in the occupational injury domain have progressively executed this NLP technique using a spectrum of machine learning (ML) and, more specifically, deep learning (DL) algorithms to improve text classification tasks (*Cheng, Kusoemo & Gosno, 2020*).

*Yedla, Kakhki & Jannesari (2020)* extracted the occupational injury narratives of the mining industry using Word2Vec, subsequently trained with several ML algorithms, and the random forest (RF) model was revealed to be the best-performing model. Similarly,

*Goldberg (2022)* compared several ML classifiers, trained with TF-IDF, Word2Vec, and Global Vector (GloVe) word embeddings, respectively to predict the outcomes of occupational injury, mainly from the construction industry. Although the ML classifiers, such as support vector machine (SVM), K-nearest neighbors (KNN), decision tree (DT), and RF were the preferred algorithms to train the occupational injury narratives (*Baker, Hallowell & Tixier, 2020b*; *Goh & Ubeynarayana, 2017*; *Sarkar et al., 2020*), the NLP-based DL techniques have been recommended to enhance text classification tasks (*Khairuddin et al., 2022*; *Zhong et al., 2020*). This is because the architectures of neural networks are better suited for capturing the complexity of language relationships due to their ability to learn hierarchical features (*Cheng, Kusoemo & Gosno, 2020*; *Young et al., 2018*). For example, *Zhang (2022)* implemented a simplified deep neural network trained with Word2Vec to classify occupational injuries, whereas *Jing et al. (2022)* developed a word-embedding DL model, namely LSTM-Word2Vec, to categorize the types of occupational injuries in the chemical industry.

Although existing research has made significant strides in integrating NLP techniques and ML algorithms into occupational injury severity prediction, there are notable gaps in our understanding of the optimal utilization of these techniques. There has been limited exploration of alternative methods for text representation, which could offer improved performance or interpretability compared to exclusively utilizing one text representation method (*Kamyab, Liu & Adjeisah, 2021*). Additionally, there is a tendency to focus on traditional machine learning algorithms rather than advanced techniques that could potentially enhance the predictive performance (*Sarkar & Maiti, 2020*). Therefore, this study distinguishes itself from the existing literature through a novel and innovative approach to text representation techniques. Unlike prior studies that have predominantly relied on a single type of text representation method, either TF-IDF, Word2Vec, or GloVe embeddings, this study proposed a comprehensive approach through fusion strategy by integrating TF-IDF and GloVe. These combinations are equipped to extract more comprehensive understanding and meaningful information, thus contributing to improved predictive performance.

In addition, this study proposes a significant emphasis on advancing state-of-the-art DL algorithms. A key differentiator is the incorporation of the Bidirectional-LSTM (Bi-LSTM) architecture, unlike traditional ML algorithms, the application of the proposed Bi-LSTM architecture in this study aligns with the nature of occupational injury narratives, in which understanding the sequential context is essential. Consequently, the development of innovative DL architectures is believed to significantly enhance text classification tasks by enabling models to better capture and memorize sequential dependencies (*Baker, Hallowell & Tixier, 2020a*; *Yedla, Kakhki & Jannesari, 2020*). In addition, by proposing an innovative fusion of text representation methods with the advanced capabilities of Bi-LSTM, this study provides a holistic and comprehensive approach for occupational injury severity prediction. This contribution aims to contribute significantly to the literature, not only by improving predictive performance, but also deepening the context of the temporal and sequential dynamics inherent in occupational injury narratives.

The primary objective of our study was to accomplish two noteworthy outcomes: First, our objective was to enhance the overall performance of the classification model by focusing on improving its predicted performance. Enhancing the dependability of occupational injury severity estimates is crucial since it offers significant insights for employers, healthcare providers, and the government. The ability to make precise forecasts facilitates an enhanced comprehension of the potential hazards linked to various forms of injuries. In addition, our research focuses on the analysis of text narratives that were extracted from injury reports. The narratives provide numerous contextual details related to each incident, encompassing particular concerning the conditions, origins, and consequences of the injuries. Thus, a more profound comprehension of the factors that contribute to the severity of injuries is attained, thereby enabling more informed decision-making in practical applications. Employers can proactively resolve underlying safety issues in the workplace, for instance, by identifying common themes or patterns in injury narratives. Therefore, customizing treatment plans according to the precise characteristics of injuries detailed in the reports, healthcare providers can enhance the quality of care provided to the patients. Furthermore, by leveraging the insights gleaned from injury narratives, government entities can formulate intervention programs that are specifically designed to diminish the incidence of particular injury types within particular industries or sectors. Subsequently, this knowledge can catalyze the implementation of more focused safety efforts and preventive measures in occupational settings.

Furthermore, the objective of our work was to improve the comprehensibility of the occupational injury severity categorization model. The interpretability of a model is of utmost importance as it enables stakeholders to have a comprehensive understanding of the key characteristics or traits that have a substantial impact on the prediction of injury severity. These insights hold significant value in the context of decision-making and risk assessment. By clearly identifying the important factors that contribute to the severity of workplace injuries, employers and safety professionals can adopt targeted interventions, training programs, or engineering controls to effectively minimize these specific risks. This, in turn, contributes to the establishment of a safer work environment.

Therefore, the notable contributions of this study are summarized as follows:

1) This study pioneered a novel approach in fusion technique by integrating two distinct text representation techniques: TF-IDF and GloVe embeddings. Unlike previous studies, which often relied on a single method, the proposed fusion methods enhance the capturing of both term importance and semantic relationships, thus improving the depth and richness of feature representation in occupational injury narratives.

2) A significant focus of this study was the development and application of the proposed Bi-LSTM architecture. Our deliberate choice of this modern DL algorithm represents a notable advancement beyond other conventional ML models, addressing the inherent limitations and improving our understanding of temporal dynamics within occupational injury narratives.

3) While other studies have focused solely on either conventional ML algorithms or modern DL models, this study conducts a comprehensive comparative analysis

encompassing both. Through this comparative work, this study aimed to provide valuable insights into the most effective technique for this specific predictive task.

4) This study also goes beyond the predictive performance by optimizing the models for enhanced interpretability and practicality through in-depth feature importance analysis and random search hyperparameter tuning. This emphasis on practical applicability distinguishes our study, thereby making it relevant and valuable for occupational safety practice. The priority of developing the optimized predictive model in this study was to recognize the significance of actionable insights in real-world occupational safety scenarios.

By integrating innovative text representation techniques, advancing deep learning capabilities, conducting a thorough comparative analysis, and optimizing models for practical applicability, this study contributes significantly to the existing literature and sets the stage for more effective and comprehensive approaches in this critical occupational safety domain.

## MATERIALS AND METHODS

### Dataset

The dataset used in this study was acquired from the United States Occupational Safety and Health Administration (US OSHA) database between January 2015 and June 2023 (https://www.osha.gov/severeinjury). An injury narrative column was selected as the primary text dataset. Each entry in this column represents a textual description of a specific workplace injury event; the circumstances, events, and factors that led to the workplace injury, providing valuable contextual information for predictive analysis in this study. Each row consists of injury narratives and their corresponding injury severity labels assigned by the trained experts, which were the safety and health personnel with the assistance of the occupational health doctor. The severity labels provided in the dataset were hospitalization and amputation.

The dataset initially consisted of 83,821 rows of textual data. Following a rigorous data-cleaning process to ensure data completeness and reliability, the dataset was refined to 83,294 rows. Data completeness was ensured by confirming that each record in the dataset contained essential information, including injury narratives and assigned occupational injury severity labels. This involved checking whether each row had an assigned severity label or not. Additionally, records lacking injury narratives were identified as incomplete and removed from the dataset. As a result, approximately 0.63% of the total records were removed because of incompleteness, resulting in a refined dataset for analysis.

### Text-preprocessing

In this study, the injury narratives underwent a series of text preprocessing steps using NLP for text standardization and to remove irrelevant information for further analysis. These procedures aimed to standardize the text and eliminate any extraneous material that could hinder subsequent analysis. The process included in this study encompassed the elimination of non-alphabetic characters, such as symbols, arithmetic digits, and

```
Before
                                                          Text
0    Three correctional facility guards were escort...
1    Employee in the Machine Shop received second d...
2    A truck driver fell approximately 4 feet while...
3    An employee's leg was pinned between a truck a...
4    An employee working on the Line 6 Auto-Beller ...
5    An employee was hospitalized after slipping an...
6    Employee working the flipping machine at the c...
7    Employee sustained burns during line tie-in op...
8    One employee was hospitalized after being stru...
9    An associate was using a knife to open a bag o...

After
0        three correctional facility guards escorting r...
1        employee machine shop received second degree b...
2        truck driver fell approximately feet descendin...
3        employee leg pinned truck powered pallet jack ...
4        employee working line auto beller reached mach...
5        employee hospitalized slipping falling working...
6        employee working flipping machine casting line...
7        employee sustained burns line tie operations o...
8        one employee hospitalized struck tire airing t...
9        associate using knife open bag chili cut finge...
```

**Figure 1** **The sample of text dataset before and after the text preprocessing.**

punctuation marks, to reduce extraneous information and emphasize the significant textual content (*Sankarasubramanian & Ganesh, 2020*). To preserve consistency in the text, all additional spaces, including trailing spaces and tabs, were eliminated (*Pahwa, Taruna & Kasliwal, 2018*). Moreover, stop words in the text, for example, "a" and "the" were also eliminated to reduce the dimensionality issue (*Lourdusamy & Abraham, 2018*). The text that had been cleaned was then tokenized and afterward processed through the processes of text representation. Figure 1 illustrates the sample of text dataset before and after the text preprocessing techniques.

## Text representation

The purpose of this step is to transform the tokenized text into a vectorized representation, which can then be utilized for training purposes in the ML and DL algorithms. The text representation techniques employed were Term Frequency-Inverse Document Frequency (TF-IDF) and the Global Vector (GloVe) word-embedding model.

TF-IDF is made up of two parts: the 'term frequency' (TF) and the 'inverse document frequency' (IDF). TF is determined by the frequencies of the terms in each report, whereas IDF is measured by how often the word or term appears in the overall text data. The formula of TF is $f_{i,w}$, where $i$ is a specific term in each document and $w$ is the number of documents. In this experiment, the data consists of $D$ documents, with $df_i$ representing the

frequencies of a term across the documents. The logarithmic inverse of a keyword, $idf_i$, is used to determine its IDF, as indicated in the following formula:

$$idf_i = \log\left(\frac{1+D}{1+df_i}\right) + 1.$$

The final TF-IDF score is then calculated using this equation:

$$tfidf_{i,w} = tf_{i,w} \times idf_i.$$

Following the vectorized word, a GloVe word embedding was performed. In this study, a pre-trained GloVe model named "Glove.6B" was applied to construct the word vector representation. This pre-trained model is a 100-dimensional vector that was trained on six billion tokens from Wikipedia articles and the Gigaword dataset. It is freely available under the terms of a Public Domain Dedication and License (*Yu et al., 2018*).

In the initial phase, this study adhered to conventional practices in unstructured text learning by executing TF-IDF for training ML models and a word embedding model for DL models. It has been stated that text vectorization, such as TF-IDF performs better in ML models (*Bharti et al., 2022*), whereas the word-embedding model, for example GloVe works well in DL models (*Kilimci & Akyokus, 2018*). This standard approach served as the baseline for the text representation experiment.

Subsequently, an innovative technique called TFIDF-GloVe was proposed to train all classifiers. This study introduced an innovative approach to text representation by integrating two techniques, which were TF-IDF and the GloVe word embedding model (TFIDF-GloVe). This combination is anticipated to enhance the text representation, thus producing a prediction model that is more precise and accurate (*Kamyab, Liu & Adjeisah, 2021*). The TFIDF-GloVe vector representation was used as input features to learn unstructured injury narratives for predictive analysis. A simplified pseudocode for this proposed text representation method is presented in Table 1.

## The proposed model

This study emphasizes the Bidirectional LSTM (Bi-LSTM) model as a revolutionary technique for enhancing occupational injury categorization based on injury narratives. The motivation behind employing Bi-LSTM architectures lies in their inherent ability to capture contextual dependencies from both the past and future contexts of each word in a sequence (*Wu et al., 2021*).

In the first stage of the model, the vectorized text representation is provided as input features to Bi-LSTM. Once the vector representations were obtained, they were channeled into an embedding layer to map each vector into a continuous space that preserves the semantic meaning. As the input sequences pass through the Bi-LSTM layers, they undergo feature extraction and representation learning. Bi-LSTM units distill the semantics and context of each word, transforming text vectors into a higher-dimensional feature space that encapsulates the underlying patterns of the narrative. This feature-rich representation captures not only linguistic characteristics but also contextual cues that are critical for discerning the severity of occupational injuries. The transformed features are then directed

**Table 1 Pseudocode of text experiment.**

| **Unstructured text analysis** |
| --- |

|     | **Begin** |
| --- | --- |
|     | Input: Occupational injury narratives (text) |
|     | Output: text representations (vector) |
| 1   | *Text Preprocessing* |
| 2   | def preprocess_text(text): |
| 3   | text = remove_non_alphabetic_characters(text) |
| 4   | text = remove_punctuation(text) |
| 5   | text = remove_extra_spaces(text) |
| 6   | text = remove_stop_words(text) |
| 7   | text = convert_to_lowercase(text) |
| 8   | return text |
| 6   | *Text Representation* |
| 7   | def generate_tfidf_glove_representation(text): |
| 8   | tfidf_vector = calculate_tfidf_vector(text) |
| 9   | glove_embedding = generate_glove_embedding(text) |
| 10  | tfidf_glove_representation = concatenate (tfidf_vector, glove_embedding) |
| 11  | return tfidf_glove_representation |
| 12  | *Classifier Training* |
| 13  | def train_classifier(X, y): |
| 14  | classifier = initialize_classifier() |
| 15  | classifier.fit(X, y) |
| 16  | return classifier |
|     | **End** |

through a dense layer, which further refines the learned representation. This layer aggregates information from the sequential context and transforms it into a format conducive to making predictions. Ultimately, a final dense layer with a sigmoid activation function generates probability scores, indicating the predicted occupational injury severity level for each narrative.

Before the model development, stratified sampling was used to divide the data into two sets, with 80% acting as the training set and the remaining 20% serving as the testing set. All of these models were developed using Python programming language, leveraging its extensive libraries and packages for ML, DL, and NLP tasks. The prediction models were developed on a laptop equipped with the following specifications: AMD Ryzen 7 3700U @ 2.30 GHz with 12 GB RAM (CPU) and RadeonTM RX Vega 10 Graphics running at 1,400 MHz (GPU).

## Hyperparameter tuning

Moreover, this study employed a meticulous process of model refinement and optimization to ensure the optimum efficiency of our developed Bi-LSTM model. A key step is hyperparameter tuning. Hyperparameters serve as configuration settings that guide

| Table 2 Optimal hyperparameters of the Bi-LSTM model. | | |
|---|---|---|
| **Hyperparameters** | **Range** | **Optimal values** |
| LSTM | 128, 256, 512 | 128 |
| Dense unit | 10, 20, 30 | 10 |
| Dropout | 0.2, 0.3, 0.4 | 0.2 |
| Batch size | 32, 64, 128 | 64 |
| Epochs | 20, 25, 30 | 25 |
| Activation | ReLu, tanh, sigmoid | tanh |
| Optimizer | Adam, SGD, RMSprop | Adam |
| Output layer | Sigmoid | |
| Loss function | Binary cross-entropy | |

how the model learns from the data and generalizes its findings. In our approach, we leverage a rigorous method known as random search cross-validation (CV) with a fold size of k = 10. This technique systematically explores various combinations of hyperparameters within predefined ranges, thereby enabling us to identify the configurations that yield the best results. The hyperparameters of our proposed deep learning predictive model include the number of LSTM units, batch size, activation function, dropout, epoch unit, and optimizers. The ranges and optimal values of the hyperparameters are presented in Table 2.

## Feature importance analysis

In addition to hyperparameter tuning, this study explored the importance of the features within our predictive model. We incorporated feature importance analysis using a Random Forest (RF) feature importance algorithm. This algorithm can be effectively adapted and applied to textual data to reveal the significance of various words in influencing occupational injury severity classification outcomes. In addition, this feature importance algorithm based on RF is recommended because the tree-ensemble model can provide information on the contribution of each feature utilized in the prediction task, including its ability to handle numerous text features (*Hwang et al., 2023*; *Wang, Yang & Luo, 2016*).

After training the RF classifier, feature importance scores were computed based on the impact of each feature on the predictive performance of the classifier. This calculation was derived from the decrease in Gini impurity (*Moore, Lyons & Gallacher, 2019*). The computed feature importance scores provide a ranking of the features in terms of their influence on the classification outcomes. Higher scores indicated greater importance. This step was performed independently from the Bi-LSTM model. Then, the important features served as input features to redevelop the Bi-LSTM model. The Bi-LSTM model was retrained using the selected features as the input and the corresponding target labels. In this context, the Bi-LSTM model utilizes selected features to learn the sequential patterns and dependencies within the data. By incorporating these important features identified by RF, the Bi-LSTM model aims to leverage the most relevant information for the prediction

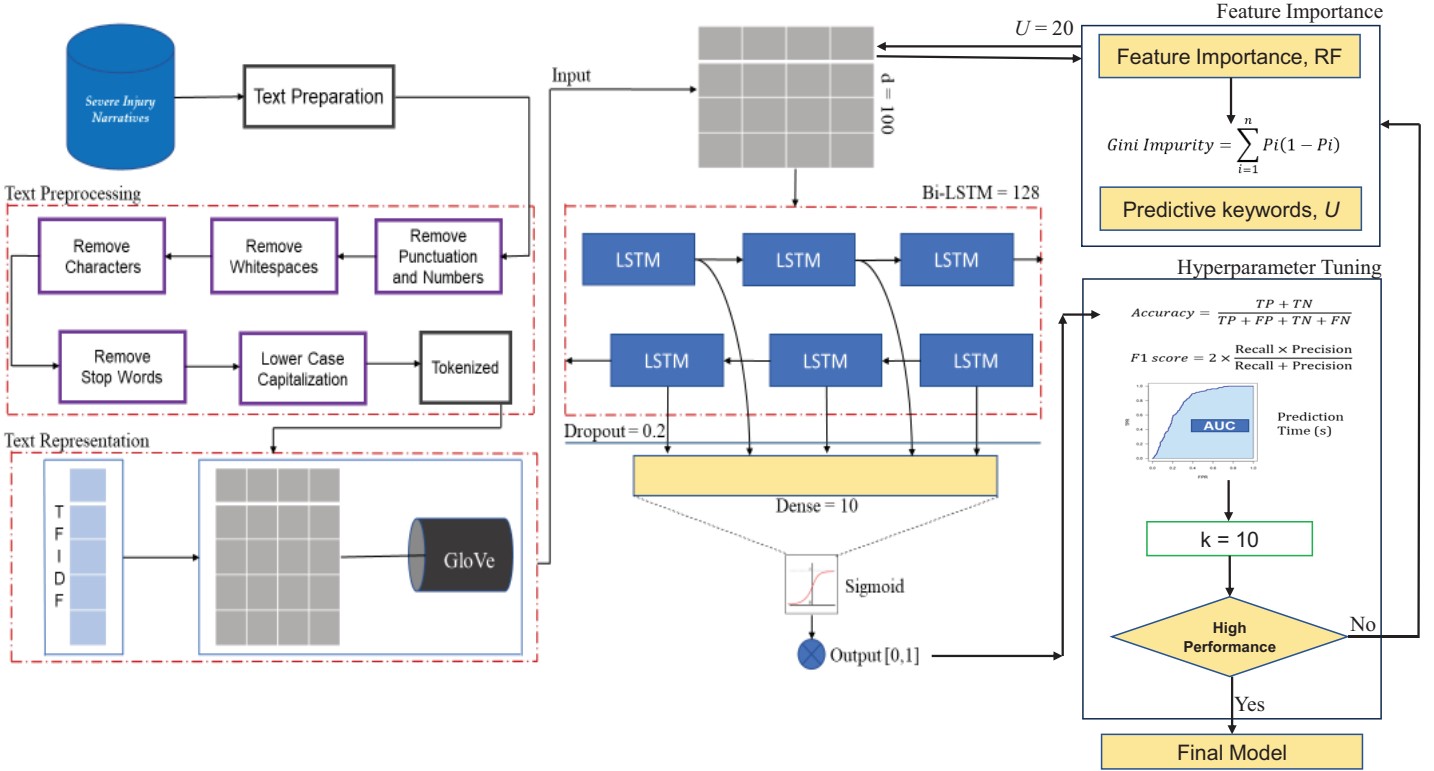

**Figure 2 The proposed framework for the optimized Bi-LSTM model.**

task and potentially achieve better predictive performance compared with using the entire feature set. To summarize, RF and Bi-LSTM serve different purposes in the predictive modeling process. RF is used for feature analysis and extraction, whereas Bi-LSTM is employed to learn sequential patterns and make predictions. The important features identified by RF serve as inputs for optimizing the Bi-LSTM model. Furthermore, this type of pipeline represents a novel exploration in predictive modeling and has been gaining traction in related studies. Such approaches have been widely adopted across various domains including clinical data classification (*Kong & Yu, 2018*; *Wu et al., 2020*), financial analysis (*Ma, Han & Fu, 2019*; *Pai & Ilango, 2020*), and solar power prediction (*Wang et al., 2023*). All these related studies acknowledge the effectiveness of combining RF for feature importance and LSTM networks for sequence modeling.

Therefore, our proposed occupational injury severity prediction model highlights the synergy between the refined Bi-LSTM architecture and model optimization steps, which harnessed the power of random search cross-validation and RF feature importance analysis. The proposed framework for the optimized Bi-LSTM is shown in Fig. 2.

## Model comparison

To comprehensively assess the efficacy of the proposed Bi-LSTM model, a thorough comparative analysis using a spectrum of established classifiers was conducted. This analytical approach provides a holistic view of the performance of the proposed model

relative to other commonly used methods. We incorporated a set of five widely used models, namely naive Bayes (NB), K-nearest neighbors (KNN), decision trees (DT), support vector machine (SVM), and random forest (RF), including long short-term memory (LSTM) and Bidirectional LSTM (Bi-LSTM).

## Evaluation metrics

In this study, standard model evaluation metrics like accuracy, precision, recall, F1-score, and AUC, were used based on a confusion matrix. The number of positive sentences that are correctly classified into the injury severity class is specified as true positive (TP), whereas, the number of negative sentences, correctly classified as negative into the injury severity class is designated as true negative (TN). The frequency of negative sentences categorized as positive is specified as false positive (FP), while the frequency of positive sentences wrongly indicated as negative in the injury severity class is classified as false negative (FN). Therefore, the aforementioned metrics were calculated based on the following equations:

$$Accuracy = \frac{TP + TN}{TP + FP + TN + FN}$$
$$Precision = \frac{TP}{TP + FP}$$
$$Recall = \frac{TP}{TP + FN}$$
$$F1 - score = 2 \times \frac{Recall \times Precision}{Recall + Precision}.$$

Additionally, the AUC metric was employed to provide a comprehensive measure of the model's ability to distinguish between different severity classes. This metric quantifies the area under the Receiver Operating Characteristic (ROC) curve. The ROC curve plots the true positive rate (TPR) against the false positive rate (FPR) across the various classification thresholds. A higher AUC value indicated better discriminatory ability.

Furthermore, the model processing times (in seconds, s) were recorded as part of the model evaluation process. This information is beneficial because it provides insights into the practicality of the model that is efficient for real-field deployment, as it requires a timely prediction capability. By incorporating a wide range of evaluation metrics and model processing times, this study provides a comprehensive comparative assessment for choosing the best-performing occupational injury severity model.

## RESULTS

This section outlines the findings of our model experimentation on text classification for predicting the severity of occupational injuries. Our analysis encompassed a binary classification task, with occupational injury severity outcomes categorized as either 'hospitalization' or 'amputation'. The prediction performances of the models were evaluated using established metrics, including accuracy, precision, recall, F1-score, AUC, and model processing times. Additionally, we present an assessment of the text

| Table 3 Comparison of text representation methods for ML models. | | | |
|---|---|---|---|
| **Models** | **Metrics** | **Text representation techniques** | |
| | | **TF-IDF** | **TFIDF-GloVe** |
| NB | Accuracy | H: 0.53 | H: **0.88** |
| | | A: 0.53 | A: **0.95** |
| | F1-score | H: 0.57 | H: **0.92** |
| | | A: 0.55 | A: **0.93** |
| KNN | Accuracy | H: 0.91 | H: **0.90** |
| | | A: 0.97 | A: **0.98** |
| | F1-score | H: 0.92 | H: **0.94** |
| | | A: **0.98** | A: 0.96 |
| DT | Accuracy | H: 0.91 | H: **0.92** |
| | | A: 0.97 | A: **0.97** |
| | F1-score | H: 0.92 | H: **0.95** |
| | | A: 0.96 | A: **0.96** |
| RF | Accuracy | H: 0.90 | H: **0.92** |
| | | A: 0.95 | A: **0.97** |
| | F1-score | H: 0.90 | H: **0.95** |
| | | A: 0.95 | A: **0.97** |
| SVM | Accuracy | H: 0.92 | H: **0.92** |
| | | A: 0.98 | A: **0.98** |
| | F1-score | H: 0.94 | H: **0.95** |
| | | A: 0.96 | A: **0.97** |

**Notes:**
The bold values mark the best performance regarding different metrics.
H, hospitalization; A, amputation.

representation techniques and interpretation of the feature importance analysis employed in this study.

## Comparison of text representation

As elaborated in the Text Representation section, five ML models were trained with TF-IDF, whereas both LSTM and Bi-LSTM models were trained with GloVe embedding to set the baseline, before the assessment proceeded with the integration of TFIDF-GloVe trained in all models. Table 3 compares the prediction performance using the accuracy and F1-score, of each ML model trained with TF-IDF alone and TFIDF-GloVe as text representation. The findings revealed that all the ML models trained with TFIDF-GloVe performed better than TF-IDF alone. Table 4 presents the performance of LSTM and Bi-LSTM using GloVe alone and TFIDF-GloVe. Based on Table 4, the findings also revealed that the integration of TFIDF-GloVe improved the prediction performance for both LSTM and Bi-LSTM.

This findings were in agreement with the previous studies done by *Kamyab, Liu & Adjeisah (2021)* and *Kilimci & Akyokus (2018)* that combined both TF-IDF and pre-trained word embedding methods to generate more accurate predictive models. They

**Table 4 Comparison of text representation methods for DL models.**

| Models | Metrics | Text representation techniques | |
| --- | --- | --- | --- |
| | | GloVe | TFIDF-GloVe |
| LSTM | Accuracy | H: 0.91 | H: **0.91** |
| | | A: 0.96 | A: **0.97** |
| | F1-score | H: **0.94** | H: 0.92 |
| | | A: 0.94 | A: **0.95** |
| Bi-LSTM | Accuracy | H: 0.91 | H: **0.93** |
| | | A: 0.96 | A: **0.98** |
| | F1-score | H: 0.92 | H: **0.93** |
| | | A: 0.95 | A: **0.98** |

**Notes:**
H, hospitalization; A, amputation.
The bold values mark the best performance regarding different metrics.

concluded that incorporating word embeddings into TF-IDF weighted vectors not only augments the feature set, but also leads to a notable enhancement in text classification tasks. This improvement stems from the capacity of the pre-trained word embedding model to capture contextual, semantic, and syntactic data within the text narratives, thereby refining the overall text representation. Next, the combination is expected to possess the ability to capture both local and global context information, including enhancing the semantic representation of occupational injury narratives (*Dogra et al., 2022*). Consequently, the integration of these two methods allows the predictive model to obtain advantages from a feature space that is both more concise and informative. Moreover, this integration effectively mitigates the problem of overfitting and enhances computational efficiency (*Lu, Ehwerhemuepha & Rakovski, 2022*) and exhibits superior performance, particularly when handling large corpora, as highlighted by *Dogra et al. (2022)*.

## Hospitalization

Table 5 provides a comprehensive overview of the performance metrics for predicting hospitalization across all models. The Optimized Bi-LSTM model achieved the highest accuracy of 0.93, whereas all the models demonstrated an impressive F1-score of 0.95, with the exception of the NB and KNN models. Moreover, the Optimized Bi-LSTM model outperformed the others in terms of AUC, with a notable score of 0.94. Our analysis also revealed that although each DL model required longer training and testing times for text representation learning, the processing times of the Optimized Bi-LSTM model were significantly improved.

## Amputation

Table 6 presents the model performance metrics for predicting the amputation. Based on the findings, the highest accuracy achieved by several models, including KNN, SVM, Bi-LSTM, and Optimized Bi-LSTM, was 0.98. However, the Optimized Bi-LSTM outperformed the other models in terms of the F1-score (0.98) and AUC (0.99). Similarly,

**Table 5 Model performance metrics for hospitalization prediction.**

| Models | Accuracy | Precision | Recall | F1-score | AUC | Training (s) | Testing (s) |
|---|---|---|---|---|---|---|---|
| Optimized Bi-LSTM | **0.95** | 0.98 | 0.94 | **0.95** | **0.94** | **997** | **62** |
| Bi-LSTM | 0.93 | 0.98 | 0.92 | 0.95 | 0.93 | 1117 | 134 |
| LSTM | 0.91 | 0.97 | 0.93 | 0.95 | 0.92 | 964 | 46 |
| NB | 0.88 | 0.96 | 0.87 | 0.92 | 0.90 | 0.013 | 0.08 |
| KNN | 0.90 | 0.94 | 0.94 | 0.94 | 0.84 | 0.16 | 3.01 |
| DT | 0.92 | 0.98 | 0.92 | 0.95 | 0.92 | 0.15 | 0.04 |
| SVM | 0.92 | 0.99 | 0.92 | 0.95 | 0.93 | 45 | 26 |
| RF | 0.92 | 0.98 | 0.92 | 0.95 | 0.92 | 3.12 | 0.12 |

Note:
The bold values mark the best performance regarding different metrics.

**Table 6 Model performance metrics for amputation prediction.**

| Models | Accuracy | Precision | Recall | F1-score | AUC | Training (s) | Testing (s) |
|---|---|---|---|---|---|---|---|
| Optimized Bi-LSTM | 0.98 | 0.97 | 0.98 | **0.98** | **0.99** | 925 | **58** |
| Bi-LSTM | 0.98 | 0.97 | 0.98 | 0.97 | 0.98 | 1019 | 95 |
| LSTM | 0.97 | 0.97 | 0.90 | 0.93 | 0.95 | 399 | 70 |
| NB | 0.95 | 0.88 | 0.98 | 0.93 | 0.96 | 0.021 | 0.01 |
| KNN | 0.98 | 0.97 | 0.96 | 0.96 | 0.97 | 0.15 | 3.13 |
| DT | 0.97 | 0.98 | 0.95 | 0.96 | 0.97 | 0.13 | 0.04 |
| SVM | 0.98 | 0.98 | 0.95 | 0.97 | 0.97 | 114 | 6.25 |
| RF | 0.97 | 0.98 | 0.95 | 0.97 | 0.97 | 2.46 | 0.13 |

Note:
The bold values mark the best performance regarding different metrics.

the Optimized Bi-LSTM model generated more efficient computational time than the other DL models.

In both prediction tasks, the Optimized Bi-LSTM models were superior to the other classifiers. They not only achieved the highest accuracy and F1-score but also exhibited exceptional discriminatory ability between prediction classes, as evidenced by the AUC. Furthermore, the optimized models showcased significant enhancements in both training and testing times, thereby highlights the efficiency gains achieved through model optimization, underscoring the practical applicability of our approach.

## Feature importance interpretation

This study identified the top 20 important keywords for both prediction tasks, as illustrated in Figs. 3 and 4. Based on Fig. 3, the presence of terms, such as "hospitalized", "fell", "caught", "machine", "blade", and "saw" suggests a focus on injuries resulting from workplace accidents involving machinery or equipment, which often lead to severe trauma requiring medical attention. References to specific body parts such as "fingertip", "finger", "thumb", "hand", and "knuckle" indicate the potential sites of occupational injury, with injuries to these affected body parts being more likely to require hospitalization due to their sensitivity and importance for functionality. Additionally, terms like "amputated",

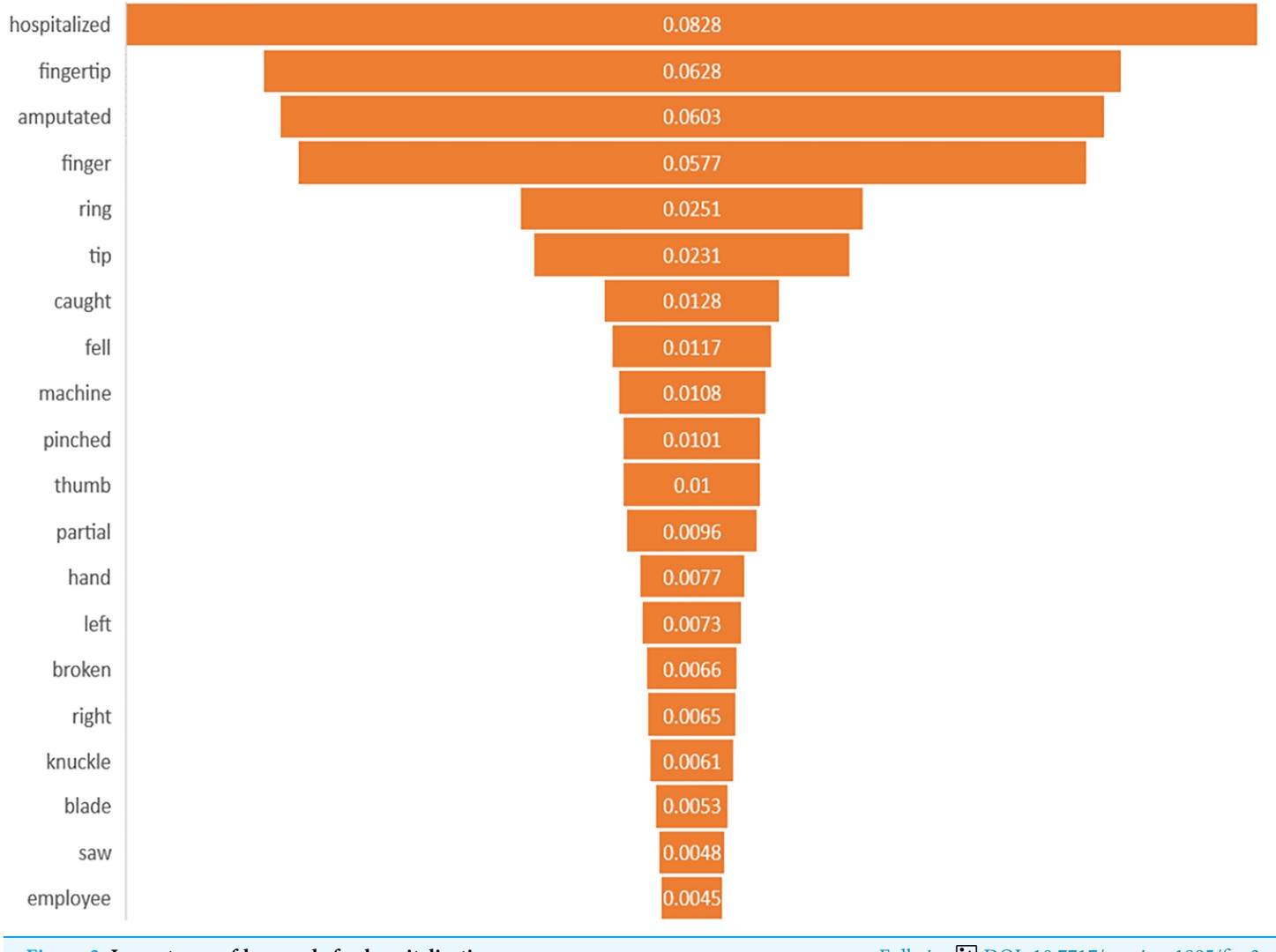

**Figure 3  Importance of keywords for hospitalization.**

"pinched", "broken", and "partial" suggest the severity of injuries, with amputations, fractures, and severe trauma increasing the likelihood of hospitalization for urgent medical care and treatment. Meanwhile, for amputation task, terms of "amputated" and "amputation" directly signify the outcome of interest, in which these keywords are crucial indicators of the injury severities being considered in the prediction model. The potential mechanism of injury lead to amputation was machinery accidents, where terms such as "machine", "blade", and "pinched" suggest the likelihood of amputation due to the high force and shard edges involved. Moreover, the inclusion of "trapped" suggests situations where body parts are confined, possibly in machinery or equipment, which can lead to severe injuries that may necessitate amputation. Based on the interpretability analysis, it is evident that the occupational injury narratives contained keywords that delineated the accident's type or causes (*Sarkar et al., 2022*), as well as the affected body parts, along with

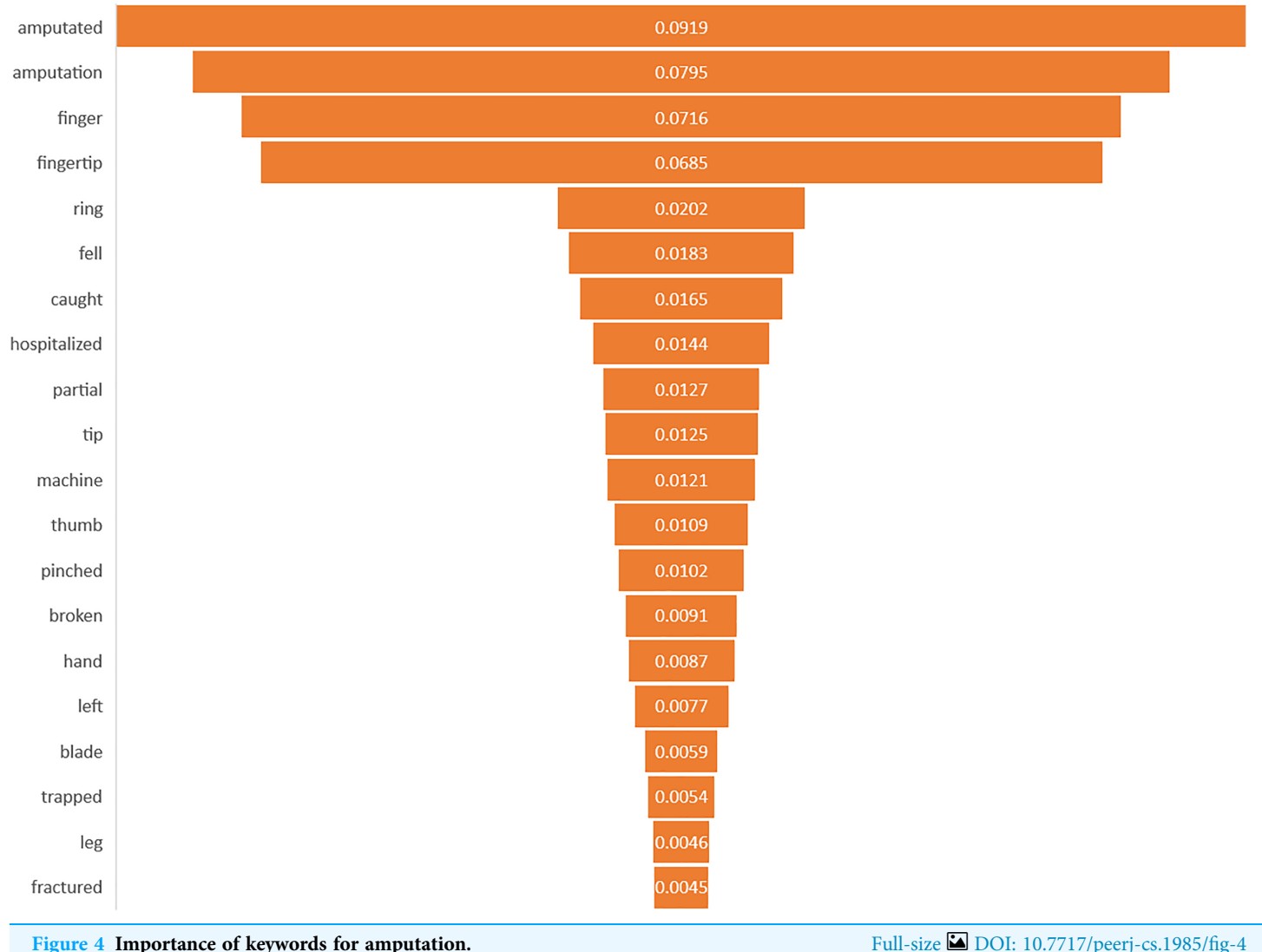

**Figure 4 Importance of keywords for amputation.**

the nature of the injury or outcomes (*Davoudi Kakhki, Freeman & Mosher, 2019*; *Kang, Koo & Ryu, 2022*; *Yedla, Kakhki & Jannesari, 2020*). These findings align with those of similar studies in the field, providing comparable insights into the predictors and consequences of occupational injuries.

By determining which features have the most significant impact on predicting occupational injury severity, the safety and health practitioner can prioritize these factors for further investigation, intervention, or preventive measures. It aids in understanding the underlying mechanisms or factors contributing to the occurrence of workplace accidents and injuries. This knowledge can significantly enhance workplace safety protocols, including the development of comprehensive job safety training programs and the implementation of rigorous occupational risk assessments in industrial settings. By doing so, it aims to effectively prevent or mitigate severe injuries that could result in

hospitalization or amputation. In summary, incorporating feature importance into the analysis of text injury narratives revealed the significance of individual features in the predictive model (*Tjoa & Guan, 2021*). It transforms the model's predictions into actionable insights and guides workplace safety initiatives and decision-making processes to effectively mitigate and prevent workplace accidents and injuries.

In terms of improving the approach, feature importance analysis helps refine predictive models by focusing on the most influential variables, thereby improving their accuracy and reliability of the predictive model. By removing less relevant or redundant features and focusing on the key predictors identified through feature importance analysis, more efficient and targeted predictive models can be developed for identifying individuals at the highest risk of occupational injuries (*Chowdhury & Turin, 2020*; *Maharana, Mondal & Nemade, 2022*).

## DISCUSSION

### Strength of the optimized Bi-LSTM model

Our findings support the Bi-LSTM model as the best classifier for text classification and are consistent with previous related studies in this field (*Girgis, Amer & Gadallah, 2018*; *Jing et al., 2022*; *Onan, 2021*; *Yang, Yu & Zhou, 2020*). In principle, the advanced structure of the proposed optimized Bi-LSTM permits a more effective flow of information in the sequential data. It contains an additional layer that can capture contextual information from both past and future sequences of words. This bidirectional nature works well with text interpretation as it enables the model to analyze dependencies that span a wider context (*Lu, Ehwerhemuepha & Rakovski, 2022*). This is beneficial for understanding complex linguistic relationships in text data. Additionally, by considering the "forward-backward" direction of context, Bi-LSTM comprehends more comprehensive text representations, thereby, playing a crucial role in making accurate predictions (*Tavakoli, 2019*).

Nevertheless, the bidirectional nature also resulted in computational demands, especially increased processing time, compared to unidirectional and simpler models (*Dogra et al., 2022*). 'Double LSTM' units themselves involve complex operations, including additional gates and layers, which require larger memory and storage capacities, thus leading to additional computational load. However, the trade-off often lies in their ability to capture more intricate relationships and context within the text data effectively.

In comparison to the unidirectional LSTM and other conventional ML techniques executed in this study, the Bi-LSTM has emerged as a robust and effective prediction model owing to its strength in providing holistic semantic representation and adaptability to diverse linguistic contexts; therefore, Bi-LSTM is an asset for enhancing superior performance in text classification. Furthermore, this comparison justifies the necessity of model optimization to improve the predictive performance of the model, including assisting in faster convergence throughout model training, thereby enabling the model to attain its optimum efficiency in a timely manner (*Ali et al., 2023*).

## Practical implications

The practical implications of our optimized predictive model extend far beyond algorithm sophistication. This model bridges the gap between applied data science and real-world industrial safety concerns. The optimized predictive model offers safety and health professionals a potent tool to foresee potential workplace incidents and injuries with higher accuracy. By analyzing historical occupational injury narratives, temporal patterns, and influential factors, the model can identify risk-prone situations and predict the likelihood of future workplace injuries. This insight empowers proactive measures, allowing organizations to implement targeted workplace safety protocols and interventions to prevent future workplace accidents.

Furthermore, the faster prediction time of the optimized model translates into quicker insights and occupational injury predictions. In industries where split-second decisions are crucial, the ability to obtain accurate predictions promptly is invaluable. In addition, a faster predictive model effectively empowers resource allocation. This is applied to deploying personal protective equipment (PPE) and engineered facilities precisely where they are most needed, thereby optimizing the utilization of resources for workplace safety preparedness.

A notable implication of our optimized Bi-LSTM predictive model lies in the cost reduction. By minimizing the severity of workplace injuries, industries can reduce their healthcare expenses, worker compensation claims, and equipment repair costs. Moreover, improved workplace safety contributes to sustained workforce productivity, thereby increasing business sustainability.

## Comparison with state-of-the-art techniques

A thorough comparison with other similar techniques utilizing data from the US Occupational Safety and Health Administration was conducted to provide insights into the relative performance and advantages of our proposed approach. In a study by *Goh & Ubeynarayana (2017)*, six ML models (NB, SVM, DT, KNN, LR, and RF) were employed to classify occupational injury outcomes, in terms of predicting the causal factors of the accident and revealed the SVM as the best performing model. Subsequently, *Zhang et al. (2019)* introduced an ensemble method that can potentially perform better than a single learning algorithm. In their study, SVM, DT, KNN, NB, and logistic regression were combined to form an ensemble model and outperformed each single algorithm in predicting the causes of the accident. Model stacking of XGBoost-RF was later introduced by *Baker, Hallowell & Tixier (2020a)* to validate the performance of the SVM and RF models in their predictive analysis. All these analyses focused only on construction injuries. Our study was in agreement that the most common state-of-the-art techniques used in text classification for occupational injury prediction were NB, SVM, DT, KNN, and RF. Although our study shares similarities with previous research in terms of the ML models employed, we contribute to the literature by providing a detailed analysis of each individual model's performance in predicting occupational injury outcomes across diverse industrial categories.

In an advanced study by *Cheng, Kusoemo & Gosno (2020)*, a deep learning approach based on NLP and gated recurrent units (GRU) was proposed. This approach utilized the GRU as the primary deep learning predictive model to predict occupational injury outcomes. The development of the model includes several similar NLP tasks, such as the removal of special characters and stop words. In terms of text representation, they exclusively employed GloVe embedding. By contrast, our approach extends beyond the utilization of GRU and GloVe embedding by comparing a wider range of ML models and text representation techniques. In addition, the setting up of the architecture in their study executed similar parameters, such as Adam activation, batch size, and dropout rate. However, the hyperparameter tuning method was not explicitly mentioned, potentially limiting the performance of their predictive model, compared to our approach that incorporated comprehensive hyperparameter tuning to optimize the performance of our proposed model. In addition to the differences outlined above, our approach aligns with *Cheng, Kusoemo & Gosno (2020)*'s suggestion to explore the use of advanced ML methods in sequential learning models such as RNN variants. This study expanded this exploration by developing other RNN variants, LSTM and Bi-LSTM with promising prediction performances.

This study builds upon the recent work by *Goldberg (2022)*, which utilized the latest and revised format of the US OSHA dataset to classify occupational injury outcomes, including predicting the likelihood of amputation and hospitalization severity. A range of ML models was similarly employed including DT, RF, SVM, NB, and Bi-LSTM for sequence modeling. Our study agrees with *Goldberg (2022)* regarding the performance of predictive models, as both studies found that the Bi-LSTM model achieved the highest performance in predicting the likelihood of amputation and hospitalization. Despite this similarity, there were notable differences in the methodologies and contributions of our study. One key difference lies in our approach to textual representation. Goldberg explored multiple word embeddings such as Word2Vec, GloVe, and BERT, whereas this study introduced the novel integration of TF-IDF-GloVe embeddings. Compared to using text representation methods alone, our study demonstrates that the novel integration of TF-IDF-GloVe embeddings yields superior performance in both prediction tasks. This hybrid approach leverages the strengths of both text representation methods to generate a more comprehensive representation of text data. Furthermore, this study goes beyond model performance evaluation to include model interpretation through feature importance analysis, a component absents in Goldberg's study. This analysis provides insights into the factors that play a crucial role in the model's predictions, enhancing the transparency, explainability and practicality of the proposed predictive models.

Therefore, this study contributes to the growing body of research on predictive modeling of occupational injury severity outcomes by incorporating workplace injury reports from a broad range of industrial sectors. Through comparisons with previous similar studies, the unique contributions and insights provided by our approach have been highlighted. In the following, we identified potential areas for future research and development based on the limitations and challenges observed in the existing techniques. By addressing these gaps, we believe that our approach, in line with previous similar

studies, can further contribute to advancing the state-of-the-art in occupational injury domain.

## Limitations and future research

Despite the promising insights and contributions provided by this study, it is essential to acknowledge certain limitations to guide future research in this domain. Although the integration of TF-IDF and GloVe embeddings offers improved text representations, other advanced techniques of language models, such as contextual embeddings, namely BERT, were not explored in this study. It is recommended that this advanced embedding model be explored, as it has been shown to achieve better performance in a wide range of NLP tasks (*Goldberg, 2022*). Next, this study primarily focused on the analysis of textual injury narratives; thus, the model's generalization to other types of data from occupational injury reports has not been explored in this context. Expanding the analysis to incorporate additional modalities such as occupational injury images or audio data from accident investigations could contribute to a more comprehensive knowledge of the severity of occupational injuries. Multimodal approaches have the potential to capture richer contextual information and improve the prediction performance (*Sarkar et al., 2022*).

Moreover, while the dataset may have limitations in terms of the number of severity levels, the availability of 'hospitalization' and 'amputation' labels still allows for meaningful analysis and insights into occupational injury severity. In practice, both hospitalization and amputation represent serious workplace accident outcomes that require immediate attention and intervention. However, this study acknowledges the limitations imposed by the binary nature of severity labels. Therefore, future research endeavors will aim to address this constraint by exploring datasets with a broader range of severity classification. Collaboration with other institutions or access to larger databases may provide opportunities to obtain datasets encompassing intermediate severity levels. An investigation of alternative data collection methods to capture a more comprehensive spectrum of occupational injury severity is proposed.

## CONCLUSION

In conclusion, our study provides valuable insights into the potential of text classification models for predicting occupational injury severity. By comprehensively comparing diverse NLP-based classification algorithms, this study makes significant contributions to enhancing workplace safety and offers a promising avenue for a precise and timely occupational injury severity prediction system. The incorporation of deep learning models, specifically our proposed Optimized Bi-LSTM models, underscores the role of advanced techniques in achieving high-performing occupational injury severity classification.

### Funding

This work is supported by the Deanship of Scientific Research, Vice Presidency for Graduate Studies and Scientific Research, King Faisal University, Saudi Arabia (Grant No.

4118). The funders had no role in study design, data collection and analysis, decision to publish, or preparation of the manuscript.

## Grant Disclosures

The following grant information was disclosed by the authors:
Deanship of Scientific Research, Vice Presidency for Graduate Studies and Scientific Research, King Faisal University, Saudi Arabia: 4118.

## Competing Interests

The authors declare that they have no competing interests.

## Author Contributions

- Mohamed Zul Fadhli Khairuddin conceived and designed the experiments, performed the experiments, analyzed the data, performed the computation work, prepared figures and/or tables, authored or reviewed drafts of the article, and approved the final draft.
- Suresh Sankaranarayanan conceived and designed the experiments, performed the experiments, analyzed the data, performed the computation work, prepared figures and/or tables, authored or reviewed drafts of the article, and approved the final draft.
- Khairunnisa Hasikin conceived and designed the experiments, performed the experiments, analyzed the data, performed the computation work, prepared figures and/or tables, authored or reviewed drafts of the article, and approved the final draft.
- Nasrul Anuar Abd Razak conceived and designed the experiments, performed the experiments, analyzed the data, performed the computation work, prepared figures and/or tables, authored or reviewed drafts of the article, and approved the final draft.
- Rosidah Omar conceived and designed the experiments, performed the experiments, analyzed the data, prepared figures and/or tables, authored or reviewed drafts of the article, and approved the final draft.

## Data Availability

The text data and code sample are available in the Supplemental Files.

## Supplemental Information

Supplemental information for this article can be found online at http://dx.doi.org/10.7717/peerj-cs.1985#supplemental-information.

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
