# Peer review of "Contextualizing injury severity from occupational accident reports using an optimized deep learning prediction model"

_PeerJ Computer Science, doi:10.7717/peerj-cs.1985_

## Round 0.1 · original submission · Major Revisions

We have completed the review of your manuscript, and after careful consideration, we believe that it holds potential. However, it requires significant revisions to meet our publication standards. We categorize this as "Major Revisions."

Please address all the suggestions and concerns raised by the reviewers in detail. One of them has attached an annotated manuscript. Each point should be considered thoughtfully, and your revisions should reflect a comprehensive enhancement of the manuscript's quality, rigor, and clarity. We encourage you to take this opportunity to strengthen your work, as a thorough revision can significantly enhance the impact and credibility of your research.

We look forward to receiving your revised manuscript and are optimistic about its potential contribution to our field.

Reviewer 2 ·

Basic reporting

Two important things are missing in the manuscript: 1. Contribution, 2. Comparison with other state of the art techniques.

Experimental design

Provide some results of text preprocessing. Only reporting accuracy will not prove the reliability of the work. Similar work is available in the literature. How it is different from them? What contribution you have done?

Validity of the findings

No comment

Additional comments

Require through revisions.

Reviewer 3 ·

Basic reporting

The author should discuss the current research progress and gaps in the Introduction section, rather than simply listing previous research.

Experimental design

This paper did not clarify the relationship between the Bi-LSTM model and the RF model. The correlation between these three modules needs to be explained.

Validity of the findings

There is a lack of model comparison and ablation study. The labels set by the author only include 0 and 1, which limits the practical application of the model. In terms of feature extraction, the author only displayed the extracted features. Are these extracted features reasonable? The correlation with existing literature has not been discussed. The insight by feature extraction is not explained.

Additional comments

This manuscript presents research about categorizing injury severity using deep learning. The author used TFIDF-GloVe for feature extraction, Bi-LSTM model architecture for classification, and RF for feature importance. However, these models are a bit traditional. The original innovation of the methodology is not clear. Meanwhile, the paper did not clarify the relationship between the Bi-LSTM model and the RF model. The correlation between these three modules needs to be explained.

In addition, there is a lack of model comparison and ablation study. The labels set by the author only include 0 and 1, which limits the practical application of the model. In terms of feature extraction, the author only displayed the extracted features. Are these extracted features reasonable? The correlation with existing literature has not been discussed. The insight by feature extraction is not explained.

Lines 41-47: The background is too long, please reduce it to 1-2 sentences.

Lines 99-100: Deep learning is a type of machine learning. The author used parallel representation of deep learning and machine learning may lead readers to misunderstand the relationship between them.

The author should discuss the current research progress and gaps in the Introduction section, rather than simply listing previous research.

Line 177: The paper presents a new text representation method TFIDF-GloVe. This method should be compared with the effectiveness of other text representation methods in the result section. Its advantages should be explained.

Lines 179-180: “Consequently, this combination is expected to produce a prediction model that is more precise and accurate, (Kamyab et al., 2021)” There is an extra comma in the sentence.

Lines 234-242: The paper uses RF to analyze the feature importance. What is the relationship between the Bi-LSTM and the RF model? What are the inputs and outputs of the RF model? The paper only shows the use of RF for feature extraction but does not explain the relationship between RF and Bi-LSTM.

Line 292: The author used AUC for performance evaluation, which was not introduced earlier

Annotated reviews are not available for download in order to protect the identity of reviewers who chose to remain anonymous.

Reviewer 4 ·

Basic reporting

The use of English in the manuscript is adequate, and the explanations provided are adequate to let the reader understand both the context in which the work is framed and the details of the model that is proposed.

The reviewer recommends checking lines 221 to 223 for a possible typo (generated instead of generates)

Please, consider using vectorized formats such as svg for the images. Currently, it is difficult to see the details of the figures, specially Figure 1, when zoomed in.

On section "Evaluation Metrics", the formula for the "Recall" is not aligned as the other formulas are.

Experimental design

The experiments and methods are well described.
However, further discussion may be interesting in relation with the following topics:

-It is stated that the data used covers the period from 2015 to 2021. However, the OSHA webpage currently shows that the latest reports correspond to the period january/2015 - may/2023. It would be interesting to update the results including this new data.

-It is stated that a "cleaning process" is carried out in the dataset, after which 66,405 rows of data are used for the work. Please, discuss what this process consists of.

-The severity labels in the dataset only contains two cases: "hospitalization" and "amputation". Do the authors have knowledge about other datasets that contains more types of severity injuries. It would be beneficial to be able to predict between intermediate severity levels.

Validity of the findings

The results presented by the authors are relevant and demonstrate a great performance of their method in this dataset.
However, the authors do not present a comparison with methods proposed by other research works on the same dataset, which makes it difficult to evaluate the real novelty of the method proposed.

If possible, include a comparison with other methods or to use other datasets which include a benchmark among similar works.

The authors indicate that the running time of the model is important for its deployment in real-world scenarios and provide the metrics for the running-time of their method. However, the same model performance may be highly dependent of the compute capabilities of the machine in which it is deployed. Please, include the hardware specifications of the machine used to perform the experiments.

Additional comments

The manuscript "Contextualizing injury severity from occupational accident reports using an optimized deep learning prediction model" presents a novel model for the prediction of the severity of work-related injuries.
The document presents a well-structured work which is relevant to the field of study.

However, this reviewer considers that some minor changes need to be done to the manuscript prior to its publication.

---

## Round 0.2 · accepted · Accept

Congratulations! All the suggestions have been addressed.

Reviewer 3 ·

Basic reporting

no comment

Experimental design

no comment

Validity of the findings

no comment

Additional comments

The authors have effectively addressed the comments raised. There are no further suggestions for modification.

Reviewer 4 ·

Basic reporting

Authors have addressed my comments and modified the manuscript accordingly.
I recommend once again that authors use vertorized graphics for the images.
I have no further comments

Experimental design

I have no further comments

Validity of the findings

I have no further comments

Additional comments

I have no further comments